**Data Availability Statement:** All data, materials, and R-scripts are available at the OSF platform (https://osf.io/8bfe2).

**Funding:** The author(s) received no specific funding for this work.

# Standing-up against moral violations: The predicting role of attribution, kinship, and severity

**David F. Urschler**[1]*, **Peter Fischer**[2], **Thomas Kessler**[3]

1 HSD Hochschule Döpfer University of Applied Sciences, Cologne, Germany, 2 Department of Social and Organizational Psychology, Institute for Psychology, University of Regensburg, Regensburg, Germany, 3 Department of Social Psychology, Institute for Psychology, University of Jena, Jena, Germany

* d.urschler@hs-doepfer.de

## Abstract

Witnesses of moral violations stand up for their moral principles, despite facing substantial costs for intervening. Notwithstanding its importance, little is known whether responsibility attributions and the relation between the victim and a witness (i.e., kinship) have different effects on the intention to intervene in situations of different severity (e.g., moral courage situations). We predict an interaction between the situation's severity and the victims' responsibility for their plight. In less-severe situations, witnesses would be less willing to help when they perceive the victim to be responsible for their plight. However, those who are not seen as responsible would receive more help. For more-severe situations, responsibility is predicted to have no effect. Opposite effects are predicted for the relationship between the helper and the victim. We further predict that perceived costs for helping mediates witnesses' willingness to intervene. Two studies showed that people are more willing to help individuals who are perceived as being innocent, but only in less-severe situations. In more-severe situations, people's willingness to intervene increases, regardless of responsibility attributions. We did not observe effects for kinship. Moreover, we provide partial evidence that witnesses of more-severe situations indeed accept higher costs for intervention.

## Introduction

People intervene on behalf of victims of moral violations despite potential high costs for themselves. So did the following three people. First, in 2001, a Greek person had been chased by a group of young right-wing skinheads and brutally beaten-up in Munich. Here, an anonymous group of young Turks helped and rescued the Greek person. They saved his life by risking their own [1]. Second, in 2014, Tuğçe Albayrak defended two young women against a verbal harassment by a group of young men in front of a fast-food restaurant. As the dispute was almost settled, one of the perpetrators tried to calm down another, still furious, perpetrator. Tuğçe was intervening in the quarrel of the former perpetrators, when the furious young man hit her with his fist. Tuğçe fell on a stone and died. Third, in 1996, a group of anti-Ku Klux Klan activists protested against a Ku Klux Klan rally. A passer-by wearing a T-shirt with a

**Competing interests:** The authors have declared that no competing interests exist.

confederate flag was attacked by the anti-Ku Klux Klan activists who knocked him down, kicked and beat him. However, one anti-Ku Klux Klan activist intervened in this attack, Keshia Thomas, an 18-year-old black woman. She used her body to protect the man until the police arrived. In these examples, people behaved courageously to stand up against norm and moral violations–a behavior that is termed moral courage [2].

Although moral courage is important for functioning societies [3–5], relatively little is known about predictors for moral courage [3, 6, 7]. This may be explained by the fact that moral courage and helping have been lumped together in one category [8, 9]. Although moral courage and helping (i.e., emergency helping) are distinct subtypes of prosocial behavior, they have been investigated interchangeably [1, 6, 10, 11]. Consequently, little is known about the predicting functions of responsibility attributions, situational characteristics (i.e., severity), and kinship for moral courage. Therefore, we first provide an overview of different prediction. Then, we examine the predicting functions of responsibility attribution and kinship on people's willingness to intervene in prosocial situations of varying severity. Additionally, we investigate an underlying psychological process (i.e., perceived costs for the helper). Thereby, we contribute to the existing literature by (a) systematically investigating the predicting functions of responsibility attributions, kinship, and severity, (b) extending these functions to moral courage situations, and (c) identifying an underlying psychological process.

## Attributional theory's prediction

Human beings have a constant pursuit of *why* [12–14]. We constantly ask ourselves why something has happened: Why did this person step on my foot? Why does an acquaintance need money? Why should I lend this lazy colleague my notes of the last team meeting? Since Fritz Heider's seminal publication [12], psychology has gained comprehensive insight into people's causal perceptions, especially, how people attribute causes for success and failure [14]. On this basis, attribution theory has been applied to predict helping [15–17]. In other words, how does the attribution of responsibility for one's situation influence helping that person?

Imagine the following scenarios: First, after sitting the whole semester in a statistic course, a classmate asks for your class notes to complete hers as she missed some sessions. You know that she missed sessions because she had been hospitalized. Second, your classmate asks for your notes because she missed sessions due to holidays. Under what condition would you be more willing to help your classmate? Attribution theory predicts that you would be more likely to help your classmate who is not responsible for her situation [16]. This pattern of results has been repeatedly observed across a variety of situations, such as helping a fallen stranger, or helping an acquaintance in a temporary financial hardship [e.g., 18]. Although previous research on attribution theory has provided significant insights regarding whether a person in need will receive help, little is known whether these effects might be observed across various types of prosocial behavior (e.g., moral courage).

## Kinship theory's predictions

In a crude evolutionary view, helping others decreases the fitness of the helping person but enhances the fitness of the receiver of help. However, people tend to help genetically related rather than genetically unrelated individuals to maximize the genetic benefit, as the genetically related individual caries the genes "for helping" with a certain probability [19]. Subsequent empirical research provided corroborative evidence for this reasoning [20–25]. This effect emerged across different helping situations [e.g., 26]. Analogous to the question about attribution theory's predictability of helping, the question arises whether these predictions generalize to various types of prosocial behavior (i.e., moral courage).

## Situational predictions

Research has shown that people are more willing to help when a situation is perceived as more severe or harmful for the person in need [27–31]. Beyond that, research revealed that responsibility attributions determined helping behavior better than kinship in less-severe situations; this pattern was reversed in more-severe, with kinship taking precedence over responsibility attributions [32, 33]. This finding implies that in certain situations attribution theory better predicts prosocial behavior than kinship and vice versa.

Similarly, protection motivation theory predicts that appraisal of a situation's severity affects people self-protection [34, 35]. In more-severe situations, people's motivation for self-protection increases. Consequently, people's appraisals of another person's plight should similarly affect people's decision to act prosocial [27]. This implies that the goal of protection overrides causal attributions for being in a plight in more-severe situations, whereas not so in low-severe situations. However, when a close relative is in a plight than the goal of protections should be amplified.

## How to distinguish moral courage from helping?

Helping is defined as well-intended voluntary assistance given to individuals [36, 37]. However, there are just emerging definitions of moral courage [6, 9–11, 38, 39]. It is characterized as a behavior, mostly borne of a minority position that is demonstrated after an intervener's subjective sense of justice is violated. Moreover, it occurs in situations where intervening is likely to have highly negative social consequences for anyone who does so [1, 6, 11, 38–40].

Situations that trigger moral courage or helping have some similarities. In both situations someone is in need, and people can feel obliged to assist others who are in need [41]. However, moral courage and helping may be distinguished by: (a) the costs of acting, (b) social constellations, (c) salience of societal norm, and (d) situation perception. As illustrated in the initial examples of this article, moral courage is sometimes associated with high personal costs of acting for the helper. Note that costs for a moral courageous act can vary dramatically. They can range from lower costs such as slight feelings of discomfort when speaking out in public and social disapproval to high costs such as being threatened or even attacked [42, 43]. Whereas, helping is mostly connected with relatively low personal costs for helping such making an emergency call [7, 11, 31, 38].

According to the arousal: cost-reward model [44], the costs for acting are a mixture of potential costs of intervention and potential costs of nonintervention. To illustrate this way of reasoning, recall Keshia Thomas' case. The potential costs for defending the member of an opposing group against one's ingroup action are obvious (i.e., being attacked by him, being socially disapproved by one's ingroup). In her case, the potential costs for non-intervention could have been feeling of guilt or being upset, and the bad feeling to "support" actions that one condemns on the other side. Note that the associated negative social consequences such as being disapproved of or excluded by ingroup members are crucial to distinct moral courage from heroism. Somebody who acts heroic can anticipate positive social consequences like applause or admiration, whereas in the immediate moral courage situation (and often also afterwards) a helper cannot necessarily expect positive outcomes. Instead the helper might be insulted, excluded or even prosecuted [8, 41, 45–47].

Moral courage differs from helping by the salience of societal norms. In moral courage situations people intervene because others violate social norms, whereas in helping situations there is only a person in distress [38, 48]. Additionally, intervening in moral courage situations can violate other social norms, but not necessarily in helping situations [38]. For example, a student publicly speaks up against a professor in a lecture, who had told an offensive joke.

Here, the student intervened due to a violation of a salient societal norm (i.e., gender equality). However, the student's intervention may have violated another salient societal norm (e.g., how to behave in lectures).

Moral courage distinguishes from helping also by people's perceptional speed [1, 7, 10]. People identify moral courage situations faster than helping situations, as these are associated with less ambiguity. Consequently, people feel more personal responsibility to intervene [1, 31]. Moreover, moral courage situations elicit moral outrage at a perpetrator due to the perception of norm violations [5, 49], which, in turn, foster intervention, but not in helping situations [7, 50]. This has been shown for self-reported moral courage [7] and as well as real world incidences of moral courage [10].

## The present research

The aim of the present research is twofold. First, it seeks to systematically investigate whether responsibility and kinship relations differently affect people's helping intentions across different types of prosocial behavior. Research on attribution theory has repeatedly shown that people are more willing to help when they perceive a person in need as being not responsible for his or her plight [e.g., 16, 17]. Although previous research provided great insight, it has, however, mainly opted to investigate attribution theory's predictability in one specific type of prosocial behavior per experiment [16, 18], and has mainly focused on less-severe situations [e.g., 16, 51, 52; for exceptions see, 28, 48]. Therefore, little is known whether these findings to generalize to moral courage scenarios. Although research on kinship theory has examined different prosocial situations, varying in severity [23, 26, 53], little is known whether the effects of kinship generalize to moral courage. Thus, the present research systematically investigates the interplay between responsibility attributions and kinship across different types of prosocial behavior (e.g., moral courage). In other words, we aim to answer the question whether predictions derived from attribution and, respectively, kinship theory can be transferred to moral courage situations. To answer this question is of theoretical importance because moral courage is a type of prosocial behavior that distinguishes from helping by several factors.

Second, the present research aims to contribute to better understanding of the underlying processes of prosocial behavior (e.g., moral courage). In line with the predictions of the arousal: cost-reward model [44, 53, 54] and previous experimental research [55] we propose perceived costs for intervening as a potential underlying psychological process. The model predicts that arousal emerges by witnessing another person's distress, which is directly related to the clarity, severity, and duration of the victim's need. When a person attributes his or her empathic arousal to the victim's distress, the increased arousal leads to an aversive state, which people tend to avoid. This aversive state can be effectively reduced by helping the person in need, whereby, people accept costs for intervening. Note that previous research has provided empirical evidence for this consecutive reasoning [55], therefore, we focus on the perceived costs for intervening as a potential underlying psychological process. Following Fiedler, Harris, and Schott's recommendations [56], we argue that perceived costs for intervening as an underlying process might be a more parsimonious approach than studying situational and dispositional predictors of prosocial behavior (e.g., moral courage) separately. Note that we do not criticize the existing work on moral courage in any case, on the contrary, we highly value these contributions. However, we argue that they could be integrated in costs for intervening.

Based on existing literature on attribution theory and on protection motivation theory, we predict an interaction between severity of situations and responsibility of victims for their plight. In less-severe situations, people's willingness to intervene is predicted to decrease, when the receiver of help is perceived to be responsible for her plight, because failing to help would

have little consequences. However, we predict that people's willingness to intervene increases when they perceive the receivers of help as being not responsible [14, 16]. In more-severe situations, we predict no significant effect for responsibility attributions, because more-severe situations are associated with a potentially high personal costs for failing to intervene and are understood to require intervention more quickly and clearly [1, 31, 32].

Furthermore, we predict an opposed interaction effect between kinship and the situation's severity: in more-severe situations, people's willingness to help a relative is higher, compared to a non-relative, but no significant effect of relationship is predicted for less-severe situations, because helping increase ultimately the prevalence of a helper's genes [23, 57].

Finally, based on the predictions of the arousal: cost-reward model and subsequent empirical support, we predict that the situation's severity will be associated with higher perceived costs for intervening. Higher perceived costs, in turn, will increase people's willingness to intervene.

## Study 1

Study 1's purpose was to investigate our predictions that people's intention to intervene increases for receivers of help that are perceived as not responsible for their plight, and decreases for responsible ones in less-severe situations. No significant effect is predicted for more-severe situations (e.g., moral courage). Additionally, Study 1 aimed to investigate our predictions that people's intention to intervene increases for receivers of help who are relatives, and decreases for non-relatives in more-severe situations, whereas no kinship effect is predicted for less-severe situations.

### Method

Note that we report all measures, manipulations, and exclusion in our studies. All relevant materials, questionnaires, data, and R-scripts are available on the OSF platform (https://osf.io/8bfe2/). All performed statistical tests are two-sided, and on an alpha-level of $p \leq .05$.

**Power analysis, participants, and design.** We based our power considerations on comparable research that revealed small ($\eta^2 = .02$) to medium ($\eta^2 = .16$) effects [32]. Consequently, we aimed to detect a small effect ($\eta^2 = .05$) with a power of .80. A power analysis using GPower [58] indicated a required sample size of 249 participants to detect a small interaction effect ($\eta^2 = .05$). We factually tested 290 participants (223 women, 65 men, two participants did not indicate their sex), to compensate for a minimum attrition rate of 10% [59].

Their age ranged from 18 to 39 years ($M = 21.23$, $SD = 2.40$). They were recruited at a university lecture and received course credit in exchange for their participation. All participants had provided written consent prior they experiment started. In line with the University of Regensburg's ethic regulations, we scrutinized our studies to assess whether a full ethical approval would be sensible. This assessment indicated that we did not need further ethical approval for our experiments.

To test our predictions, a 2 by 3 by 2 between-subjects design was utilized, permitting the manipulation of a receiver of help responsibility for her plight ("responsible" vs. "not responsible"); the situation's severity ("less-severe" vs. "moral courage" vs. "more-severe"); and the kinship between the help-giver and the person in need ("sibling" vs. "acquaintance"). We used R's sample()-function to randomly assign participants to one of the twelve experimental conditions [60]. The R-code can be retrieved from our OSF page.

**Procedure.** At the beginning of the study, all participants were informed that their data will be stored anonymously and that they could withdraw at any time without giving a reason. Subsequent, the participants were asked to read a short vignette, which portrayed a person

who needed help. They then answered a questionnaire that asked them to evaluate the situation and disclose their behavioral intentions. Finally, they were fully debriefed about the study's purpose.

**Scenarios.** The less-severe helping and more-severe helping scenarios employed in the study were replicated from [32]. We added a vignette depicting a moral courage scenario. The participants had to imagine the following situation: an individual who did not look foreign (either an acquaintance or a sibling) was being attacked by three young men in a pedestrian zone. The perpetrators were clearly identifiable as right-wing troublemakers. In the *responsible* condition, it was stated that the person in need had verbally offended the perpetrators prior to the attack, while in the *not responsible* condition, no such information was mentioned. Note that a pretest of our materials showed that people indeed perceive more-severe situations as more severe than less-severe situations (for further details see https://osf.io/8bfe2/).

**Dependent variables.** Once they had finished reading the vignette, participants completed a short questionnaire. The questionnaire consisted of two scales, which assessed the person in need's responsibility for their plight (three items; α = .86; all presented α-values are Cronbach α-values) and the participant's willingness to intervene (three items; α = .88).

## Results

**Manipulation check and check for interfering effects.** A *t*-test revealed that participants in the high responsibility conditions perceived the persons in need as being significantly more responsible for their plights ($M = 4.01$, $SD = 1.86$) than participants in the low responsibility conditions ($M = 1.62$, $SD = 1.09$), $t(288) = 13.38$, $p < .001$, $d = 1.56$, 95% CI [1.30, 1.83]. For transparency, we recalculated the effect size of previous research [32] that revealed a Cohen's $d = 2.63$, hence we state that we found a smaller effect size. Nevertheless, we may conclude that the responsibility manipulation was successful.

Previous research revealed that men are more likely to help than women [61]. Consequently, we performed a log-linear analysis to check whether participants were equally distributed across conditions regarding their gender. The model indicated no significant difference concerning participants' gender distribution across conditions, $\chi^2 (29) = 13.99$, $p = .991$.

Moreover, the existing literature provides, although somewhat mixed, evidence that age can predict people's willingness to intervene. Therefore, we ran a linear regression to check whether participants' age predicted their willingness to intervene. The results indicate that age did not significantly predict the willingness to intervene, $F(284) = 0.30$, $p = .581$.

**Effects of attribution, situational attributes, and kinship on helping.** A 2 by 3 by 2 ANOVA revealed significant main effects for both *responsibility*, $F(1,278) = 8.59$, $p = .004$, $\omega^2 = .02$, and *severity*, $F(2,278) = 14.96$, $p < .001$, $\omega^2 = .08$, as well as a marginal effect for *kinship*, $F(1,278) = 3.58$, $p = .059$, $\omega^2 = .01$. See Table 1 for means, standard deviations, and cell sizes.

**Table 1. Means and standard deviations of participants' willingness to intervene ratings (Study 1).**

| | Kinship | | | | | | | | | | | |
| | Sibling | | | | | | Acquaintance | | | | | |
| | Responsible | | | Not responsible | | | Responsible | | | Not responsible | | |
| Situation | M | SD | n | M | SD | n | M | SD | n | M | SD | n |
|---|---|---|---|---|---|---|---|---|---|---|---|---|
| Less-severe | 5.16 | 1.66 | 23 | 5.97 | 0.88 | 24 | 4.23 | 2.00 | 24 | 6.00 | 0.88 | 25 |
| Moral courage | 5.76 | 1.23 | 24 | 5.92 | 1.12 | 25 | 5.07 | 1.58 | 25 | 5.43 | 1.27 | 24 |
| More-severe | 6.35 | 0.81 | 25 | 6.04 | 1.22 | 23 | 6.48 | 0.63 | 25 | 6.30 | 1.19 | 23 |

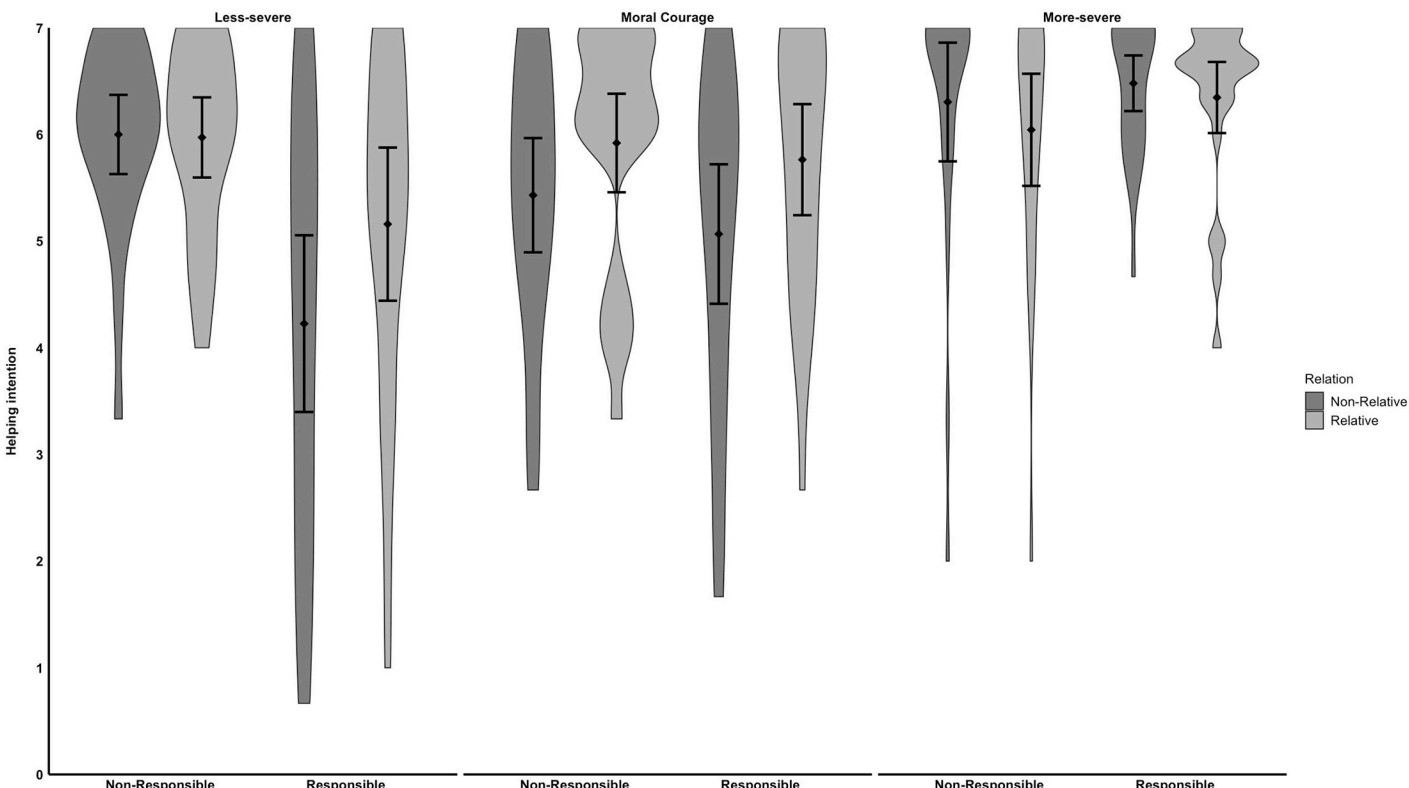

**Fig 1. Behavioral intentions to help a person in need.** Error bars represent 95% confidence intervals of the mean. We calculated the confidence intervals as recommended by [62].

The ANOVA also revealed a significant interaction between *Responsibility* and situation's *Severity*, $F(2,278) = 9.06$, $p < .001$, $\omega^2 = .05$. A simple effect analysis revealed that participants' willingness to intervene increased when a receiver of help was perceived as not responsible for the plight, $M = 5.99$, $SD = 0.88$, compared to a responsible one, $M = 4.67$, $SD = 1.89$, $t(278) = 4.98$, $p < .001$. However, responsibility did not significantly affect people's willingness to intervene in the moral courage situation (not responsible: $M = 5.68$, $SD = 1.21$; responsible: $M = 5.41$, $SD = 1.45$), $t(278) = 1.02$, $p = .312$. Similar results could be overserved for the more-severe situation (not responsible: $M = 6.17$, $SD = 1.25$; responsible: $M = 6.41$, $SD = 0.72$, $t(278) = -0.91$, $p = .358$).

Neither the *Severity* by *Kinship* interaction, $F(1, 278) = 2.65$, $p = .073$, nor the *Severity* by *Responsibility* interaction, $F(1, 278) = 2.10$, $p = .150$, was significant (see Fig 1). Although we did not predict a *Severity* by *Responsibility* by *Kinship* interaction, we included the three-way test for transparency. The three-way-interaction was not significant, $F(2, 278) = 0.78$, $p = .457$.

## Discussion

As predicted, Study 1 revealed that people's intention to intervene is affected by responsibility attributions in less-severe situations, but not in more-severe situations (e.g., moral courage). Participants' intention to intervene increased for receivers of help that were perceived as not responsible for their plight, and decreased for responsible ones in less-severe situations. However, no responsibility-based effects could be observed for more-severe situations (e.g., moral courage), thus indicating that attribution theory is more suitable to predict prosocial behavior

in less-severe situations. Contrary to our predictions, Study 1 did not provide support for the notion that kinship relations affect people's willingness to intervene. Neither did our results reveal a main effect for kinship nor an interaction with the situation's severity. One potential explanation for this result and limitation of Study 1 might be that we did not check whether all participants had siblings, which may have caused response problems for only children assigned to a sibling condition. Therefore, we changed to the broader term 'close relative' in Study 2. Some might argue that we have used only one moral courage scenarios so far. We thus opted to employ a different moral courage scenario in Study 2 to increase the generalizability of our findings. Finally, Study 1 is unable to address the question of which psychological processes may drive these findings. Study 2 thus aims to provide an answer to this question.

## Study 2

Study 2 aimed to replicate Study 1, and to examine costs for helping as an underlying psychological process. As in Study 1, we hypothesized that people's intention to intervene increases for receivers of help that are perceived as not responsible for their plight, and decreases for responsible ones in less-severe situations. No significant effect is predicted for more-severe situations (e.g., moral courage). We also hypothesized that people's intention to intervene increases for receivers of help who are relatives, and decreases for non-relatives in more-severe situations, whereas no kinship effect is predicted for less-severe situations.

Additionally, Study 2 aimed to clarify the underlying psychological process that might explain why people's willingness to intervene increases as the perceived severity of a situation rises. Based on the arousal: cost-reward model [44], we hypothesized that the perceived costs of intervening could be an underlying psychological process. More precisely, the arousal: cost-reward model proposes that people witnessing another person's distress feel a certain level of arousal, and that this arousal increases as a situation becomes more severe. These increases in arousal become increasingly unpleasant for the individual experiencing them, thus motivating behavior to reduce the arousal. An effective way to reduce negative arousal is to help the person in need, and thus decrease their level of distress [55].

### Method

**Power analysis, participants, design, and procedure.** We planned our sample size a-priori based on the same power considerations as in Study 1. A power analysis indicated a required sample size of 315 participants to detect a small interaction effect ($\eta^2$ = .05). We factually tested 346 participants (203 women), factually tested 290 participants (223 women, 65 men, two participants did not indicate their sex), to compensate for a minimum attrition rate of 10% [59].

Their age ranged from 18 to 77 years ($M$ = 30.63, $SD$ = 12.05). Participants were recruited via e-mail lists and received no payment for participating. All participants had provided written consent prior they experiment started. According to University of Regensburg's ethic committee, no ethical approval of the experiment was needed.

To test our predictions, a 2 by 3 by 2 between-subjects design was employed. We manipulated the persons in need's responsibility for their plight ("responsible" vs. "not responsible"), the severity ("less-severe helping" vs. "moral courage" vs. "more-severe"), and the kinship between the help-giver and the receiver of help ("close relative" vs. "acquaintance"). As in Study 1, we used R's sample()-function to randomly assign participants to one of the 12 experimental conditions [60]. The R-code can be retrieved from our OSF page. Study 2's procedure was identical to Study 1's.

**Scenarios.** In the less-severe and more-severe conditions, we used the same scenarios as in Study 1. However, a different scenario was employed for the moral courage condition. The participants had to imagine that they had arranged an evening meeting at a bar with a person (either an acquaintance or a close relative). After some conversation, they had visited the bathroom, and returned to find a stranger verbally sexually harassing their acquaintance /close relative. In the *responsible* condition, it was stated that the person in need had been flirting intensively with the stranger prior to the harassment, while in the *not responsible* condition, no such information was mentioned.

**Dependent variables.** We used the same items as in Study 1 to assess the responsibility of the receivers of help for their plight (three items; α = .83) and participant's willingness to intervene (three items; α = .81). Additionally, we measured the perceived costs (to the participant) of helping, and the perceived costs (to the receiver of help) of not helping using two items, based on Fischer et al. (2006): "I would not mind the costs of helping", and "Seeing the person in need, I was greatly bothered by the costs of helping" (reverse coded). Given that the two items were highly correlated (Spearman's ρ = .63), we collapsed them into a simple, single score of 'perceived costs'.

## Results

**Manipulation check and check for interfering effects.** A *t*-test revealed that participants in the high responsibility conditions perceived the persons in need as being significantly more responsible (*M* = 4.40, *SD* = 1.67) than participants in the lower responsibility conditions, (*M* = 2.04, *SD* = 1.33) $t(344) = 14.47$, $p < .001$, $d = 1.56$, 95% CI [1.32, 1.80]. For transparency, we recalculated the effect size of previous research [32] that revealed a Cohen's *d* = 2.63, hence we state that we found a smaller effect size. Nevertheless, we may conclude that the responsibility manipulation was successful.

As in Study 1, we performed a log-linear analysis to check whether participants were equally distributed across conditions regarding their gender. The model indicated no significant difference concerning participants' gender distribution across conditions, $\chi^2 (18) = 11.52$, $p = .871$. Analogously, we ran a linear regression to check whether participants' age predicted their willingness to intervene. The results indicate that age did not significantly predict the willingness to intervene, $F(344) = 0.48$, $p = .490$.

**Effects of attribution, situational attributes, and kinship on helping.** A 2 by 3 by 2 ANOVA revealed a significant main effect for *severity*, $F(2,334) = 27.65$, $p < .001$, $\omega^2 = .13$. The ANOVA revealed neither a significant main effect for *kinship* ($F(1, 334) = 0.10$, $p = .752$, $\omega^2 < .01$), nor a significant main effect for *responsibility* ($F(1, 334) = 1.78$, $p = .183$, $\omega^2 < .01$).

The ANOVA also indicated a significant interaction between *responsibility* and *severity*, $F(2, 334) = 5.84$, $p = .003$, $\omega^2 = .02$. A simple effect analysis revealed that in the less-severe situation, participants were more willing to intervene when the receiver of help was not responsible for her plight (*M* = 5.33, *SD* = 1.20) compared to being responsible (*M* = 4.60, *SD* = 1.41), $t(334) = 3.45$, $p < .001$. However, responsibility did not significantly affect people's willingness to intervene in the moral courage situation (not responsible: *M* = 5.72, *SD* = 1.12; responsible: *M* = 5.97, *SD* = 0.97), $t(334) = -1.21$, $p = .227$). Similar results could be overserved the for more-severe situation (not responsible: *M* = 6.02, *SD* = 1.00; responsible: *M* = 6.02, *SD* = 1.12, $t(334) < 0.01$, $p = .999$).

Neither the *Severity* by *Kinship* interaction, $F(1, 334) = 0.85$, $p = .427$, nor the *Responsibility* by *Kinship* interaction, $F(1, 334) = 1.01$, $p = .316$, was significant. Although we did not predict a *Severity* by *Responsibility* by *Kinship* interaction, we included the three-way test for

**Table 2. Means and standard deviations of participants' willingness to intervene ratings (Study 2).**

| | Kinship | | | | | | | | | | | |
| --- | --- | --- | --- | --- | --- | --- | --- | --- | --- | --- | --- | --- |
| | Close relative | | | | | | Acquaintance | | | | | |
| | Responsible | | | Not responsible | | | Responsible | | | Not responsible | | |
| Situation | M | SD | n | M | SD | n | M | SD | n | M | SD | n |
| Less-severe | 4.72 | 1.37 | 26 | 5.06 | 1.12 | 30 | 4.50 | 1.46 | 30 | 5.62 | 1.10 | 29 |
| Moral courage | 6.01 | 1.09 | 30 | 5.88 | 1.20 | 30 | 5.92 | 0.84 | 30 | 5.45 | 1.02 | 30 |
| More-severe | 6.10 | 0.97 | 31 | 6.01 | 1.06 | 25 | 5.92 | 1.04 | 25 | 6.02 | 0.97 | 30 |

transparency. The three-way-interaction was not significant, $F(2, 334) = 1.49$, $p = .227$. For means, standard deviations, and cell sizes, see Table 2 and Fig 2.

**Mediation analysis.** Based on the arousal: cost-reward model, we tested whether the perceived costs of helping a person in need would mediate the effect of the severity of the situation on willingness to intervene. Given that our predictor is a multi-categorical variable, we conducted a treatment coded (less-severe vs. moral courage and less-severe vs. more-severe, respectively) mediation analysis as recommended by Hayes and Preacher [63].

The mediation analysis revealed that the severity of the situation indirectly influenced willingness to intervene through its effect on the perceived costs of helping a person in need. As can be seen in Fig 3, participants in the moral courage condition perceived higher costs for

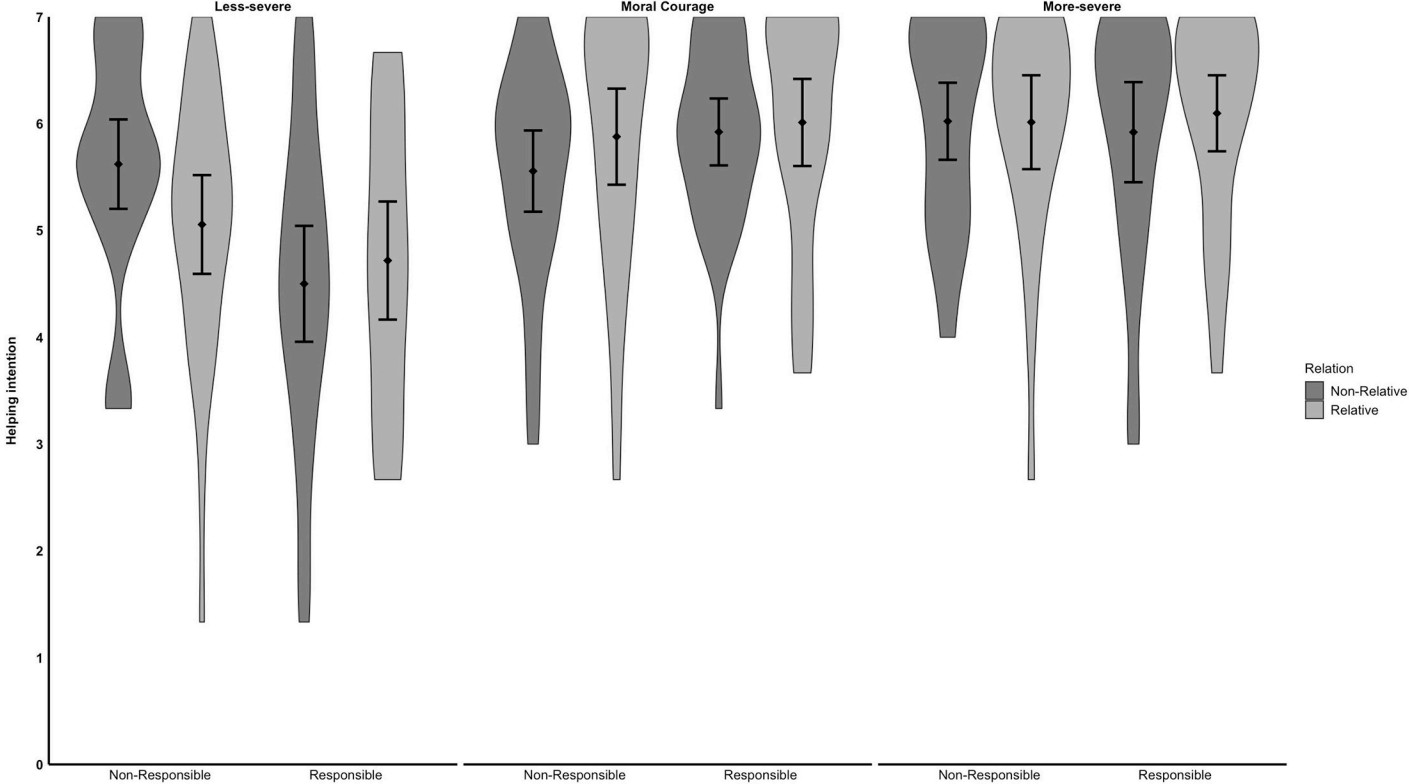

**Fig 2. Behavioral intentions to help a person in need.** Error bars represent 95% confidence intervals of the mean. We calculated the confidence intervals as recommended by [62].

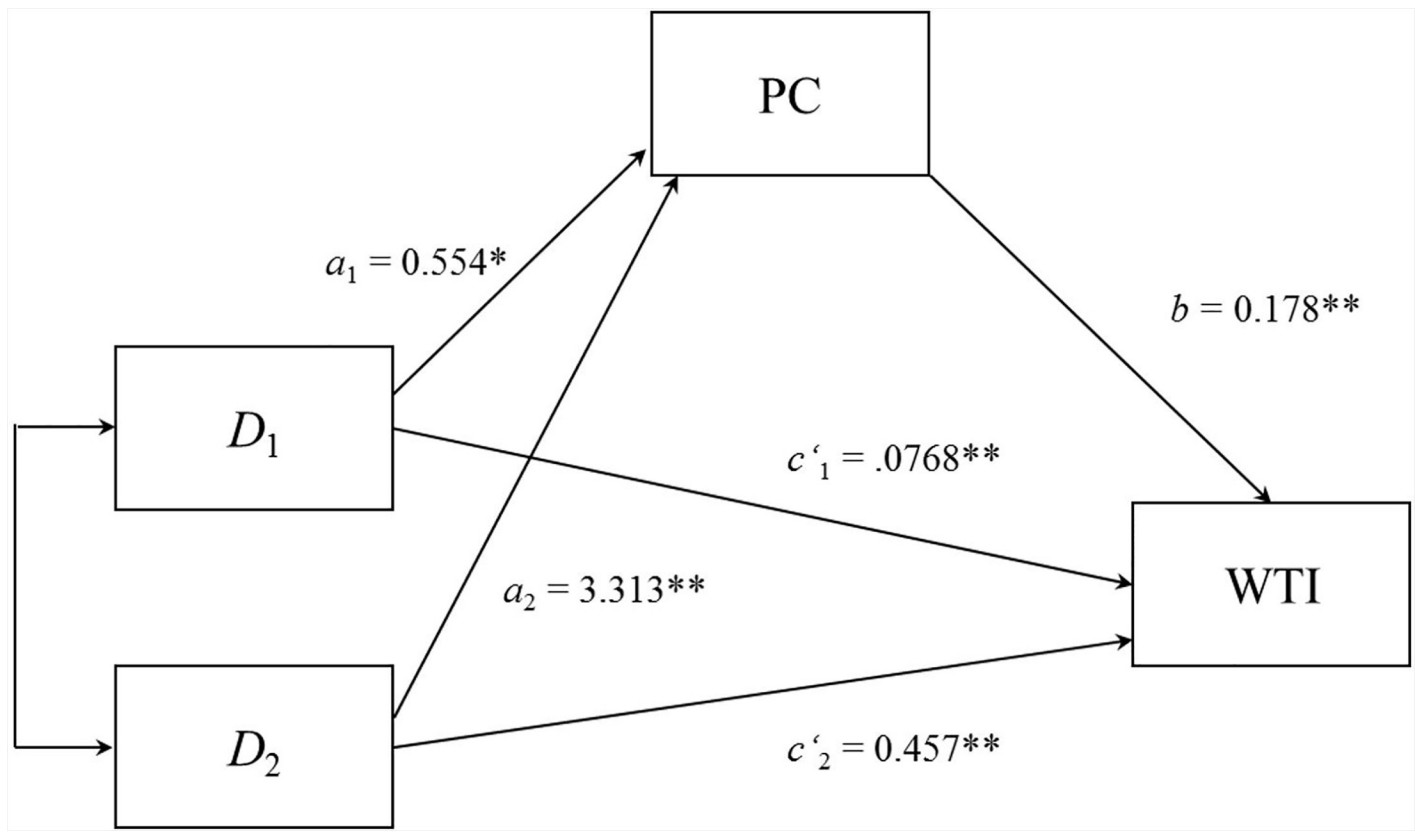

**Fig 3. The perceived costs for intervening did partly mediate the effect of the situation on the willingness to intervene.** PC = perceived costs for intervening, $D_1$ = difference between the less-severe and moral courage situation; $D_2$ = difference between less-severe and more-severe situation, WTI = willingness to intervene. $^*p < .05$ $^{**}p < .001$.

helping a person in need compared to participants in the less-severe condition ($a_1 = 0.554$). Moreover, participants in the more-severe condition perceived higher costs for helping a person in need compared to the less-severe helping condition ($a_2 = 3.313$). Participants who perceived higher costs for helping a person in need showed a greater willingness to intervene ($b = 0.178$). Bias-corrected bootstrap confidence intervals for the first ($a_1b = 0.098$) and the second ($a_2b = 0.588$) relative indirect effect based on 10,000 bootstrap samples were entirely above zero (0.022 to 0.217, 0.305 to 0.900, respectively).

However, the mediation analysis showed that costs for helping partially mediated participants' willingness to help. In other words, the situation itself predicted the willingness to intervene. Precisely, participants in the moral courage condition were more willing to intervene compared to those in the less-severe helping condition ($c'_1 = .0768$). Moreover, participants in the more-severe condition were more willing to intervene compared to the less-severe helping condition ($c'_2 = 0.457$).

## Discussion

In line with the findings of Study 1, Study 2 revealed that people's intention to intervene is affected by responsibility attributions in less-severe situations, but not in more-severe situations (e.g., moral courage). Participants' intention to intervene increased for receivers of help that were perceived as not responsible for their plight, and decreased for responsible ones in

less-severe situations. However, no responsibility-based effects could be observed for more-severe situations (e.g., moral courage). Thus, Study 2 provides further evidence for the notion that attribution theory's prediction mainly applies to less-severe situations, but not to more-severe situations. Again, in line with Study 1, but contrary to our predictions, Study 2 did not provide support for the notion that kinship relations affect people's willingness to intervene. Study 2 did neither reveal a main effect for kinship nor an interaction with the situation's severity. Additionally, the results of Study 2 revealed that people's willingness to intervene is partly driven by the perceived costs for intervening. This finding is in line with the predictions of the arousal: cost-reward model and previous experimental evidence [31, 44, 55].

## General discussion

Our results consistently showed that people's intention to intervene can be predicted attribution theory in less-severe situations, but not in more-severe, such as moral courage, situations. Participants' willingness to intervene increased for receivers of help that were perceived as not responsible for their plight, and decreased for responsible ones in less-severe situations. However, no responsibility-based effects could be observed for more-severe situations (i.e., moral courage). To be clear, moral courage indicates a higher level of severity. From a theoretical point of view, this finding implies an important qualification to predictions from attribution theory concerning people's willingness to help others. Although our findings provide further evidence for attribution theory's prediction that a person who is not responsible for his or her plight will receive more help in less-severe situations. However, in line with previous research [32], we demonstrated that attribution theory has less explanatory power when examining more-severe situations. Additionally, we extend this insight to moral courage situations.

This pattern of results might be explained by moral values, which can predict attitudes toward victims and perpetrators. For example, a victim of sexual harassment will not be perceived as responsible and blameworthy, independent of the experimental manipulation. On the contrary, people will condemn the perpetrator for acting morally wrong, and, thus act morally courageous [64]. Nonetheless, we might argue that our research is the first to demonstrate that this qualification of attribution theory extends to moral courage situations. This indicates that people's willingness to intervene is not affected by responsibility attributions in moral courage situations, despite that a person who acts morally courageous can expect negative consequences such as being insulted or socially excluded [1, 11, 45, 65].

Thus, we argue that attribution theory should be complemented by protection motivation theory's prediction that appraisal of a situation's severity affects people self-protection [34, 35]. In more-severe situations, people's motivation for self-protection increases. Consequently, people's appraisals of another person's plight similarly affect people's decision act prosocial [27]. This implies that the goal of protection may overrides causal attributions for being in a plight in more-severe situations (i.e., moral courage), whereas not so in less-severe situations. Nevertheless, we are aware of that moral courage situations vary in severity. Therefore, severity might interfere with the specific characteristics of moral courage situations. However, the pattern of results remained stable across our studies. Given that the used moral courage situations might be more severe than others, we highly encourage upcoming research to further explore whether our findings expand to less-severe moral courage situations [9].

Contrary to our predictions that people should be more willing to help a close relative than an acquaintance, especially in more-severe situations and not in less-severe situations, our results revealed no evidence for kinship theory as a predictor for the willingness to intervene. We provide five potential explanations for these differences to the existing literature [35, 43, 45–48, 56, 71–73].

First, our studies slightly differ in assessing the willingness to intervene. We constantly assessed behavioral intentions, whereas previous research assessed frequencies of prosocial behavior for less-severe situations and behavioral intentions for more-severe situations [23, 57]. Additionally, we assessed behavior intention for single-target scenarios, whereas previous research applied forced choice scenarios to examine kinship effects on prosocial behavior [26, 66–68]. Second, we manipulated kinship rather abstract, whereas previous research used specific targets [67]. To be precise, we only used close relative vs. acquaintance as categories, whereas previous studies [26, 67] used concrete exemplars for close relatives (i.e., mother, father, sister, and brother). Consequently, we cannot rule out that participants may thought of moderately close relatives (i.e., aunt, uncle, grandmother, grandfather, niece, and nephew) as close ones. Third, we argue that people may have perceived those in need as co-nationals, as indicated by kinship or acquaintance relation [22, 69], which in turn increases the willingness to help them as ingroup members [70, 71]. For example, participants could had thought of an acquaintance who supports the same sports team or of whom who attended the same concert. Therefore, they might have perceived an acquaintance as in ingroup member. This argument is strengthened by the finding that prosocial behavior increases as a function of category inclusiveness [23, 57, 72]. Fourth, although we did not assess emotional closeness, emotional closeness has been shown to account for a substantial portion of the effect of genetic relatedness on the willingness to act prosocial [67]. Thus, we encourage future research to assess emotional closeness to disentangle kinship from social closeness effects. Finally, our results might indicate that kin selected altruism and reciprocal altruism are rather intertwined than separate concepts [73, 74].

Moreover, the present research contributes to clarify underlying psychological processes in more-severe situations (e.g., moral courage). As predicted by the arousal: cost-reward model [44] and subsequent empirical support [55], mediation analysis revealed that as a situation becomes more severe, people accept higher costs for helping, which, in turn, increase people's willingness to intervene. Hence, perceived costs for helping are a mediator between severity of the situation and the willingness to intervene. However, our results cannot rule out the potential mediating effect of costs for not helping. For example, you observe a toddler locked in a car during summer heat. In this scenario, your costs for acting are rather low (e.g., calling 911, or smashing a side window), while the costs for not helping are extremely high. Thus, we highly encourage future research to examine costs for not helping as an additional underlying psychological process. We want to thank an anonymous reviewer for this important point.

Notwithstanding the present research's contribution to the existing knowledge, limitations are present. Although of high predictive value [75], we acknowledge that we assessed behavioral intentions, and not real behavior. Therefore, future research should try to replicate these findings with measurements that are closer to real behavior. Second, we only varied the moral courage scenarios; hence our sample of prosocial situations was rather limited. Moreover, we are aware that prosocial behavior in moral courage is complex and can occur in different facets. Especially, moral courage scenarios do vary in severity [9]. For example, speaking-up against an inappropriate joke might be less severe for an intervener than preventing someone from a physical attack (Study 1) or sexual harassment (Study 2), although more severe than lending class notes, or picking up coins, etc. Thus, following the Brunswikian idea of systematic and representative research designs [76–78], we highly encourage future research to systematically vary situations to get a full-fledged understanding of the predictors of prosocial behavior (i.e., moral courage). Particularly, upcoming research should orthogonally manipulate severity in moral courage scenarios. Third, some might argue that our samples differ concerning their demographics (i.e., age and gender), and that they are not representative. Although they might differ, we did not find a significant difference in their overall willingness

to intervene, $t(634) = 1.08$, $p = .282$, $d = 0.09$, 95% CI [-0.07, 0.24]. We conducted our research with only with WEIRD samples [79], hence our findings are limited in this way. Thus, we encourage future research to replicate our findings with non-WEIRD samples.

To conclude, our research shows that the willingness to act prosocial cannot be solely predicted by either attribution theory or kinship. Instead, our research shows that certain theories are more suitable to predict prosocial behavior in certain situations such as attribution theory in less-severe situations. Moreover, we demonstrate that people are willing to act morally courageous, despite anticipated costs for doing so. Finally, we show that perceived costs for helping did mediate intervening in moral courage and more-severe situations showing that people do, indeed, accept increased costs for intervening in more-severe situations. In addition to our theoretical contributions, our findings can be applied to improve training programs that foster prosocial behavior. Especially, our results can help to develop tailored exercises that aim to improve moral courage.

## Author Contributions

**Conceptualization:** David F. Urschler, Peter Fischer.

**Data curation:** David F. Urschler.

**Formal analysis:** David F. Urschler.

**Methodology:** David F. Urschler.

**Writing – original draft:** David F. Urschler.

**Writing – review & editing:** David F. Urschler, Thomas Kessler.

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
