## [Decision Letter · Decision Letter 0]

11 Jul 2023

PONE-D-23-02496Standing-up against moral violations: The predicting role of attribution, kinship, and severity.PLOS ONE

Dear Dr. Urschler,

Thank you for submitting your manuscript to PLOS ONE. After careful consideration, we feel that it has merit but does not fully meet PLOS ONE’s publication criteria as it currently stands. Therefore, we invite you to submit a revised version of the manuscript that addresses the points raised during the review process.

 Please submit your revised manuscript by Aug 25 2023 11:59PM. If you will need more time than this to complete your revisions, please reply to this message or contact the journal office at plosone@plos.org. Please include the following items when submitting your revised manuscript:A rebuttal letter that responds to each point raised by the academic editor and reviewer(s). You should upload this letter as a separate file labeled 'Response to Reviewers'.A marked-up copy of your manuscript that highlights changes made to the original version. You should upload this as a separate file labeled 'Revised Manuscript with Track Changes'.An unmarked version of your revised paper without tracked changes. You should upload this as a separate file labeled 'Manuscript'.

We look forward to receiving your revised manuscript.

Kind regards,

Yasir Ahmad

Academic Editor

PLOS ONE

Journal Requirements:

2. Please change "female” or "male" to "woman” or "man" as appropriate, when used as a noun (see for instance https://apastyle.apa.org/style-grammar-guidelines/bias-free-language/gender).

**Additional Editor Comments:**

The reviewers have provided very detailed comments which will provide great help for preparing the revision of your manuscript. Kindly address all the major concerns of the reviewers.

Reviewers' comments:

Reviewer's Responses to Questions

**Comments to the Author**

1. Is the manuscript technically sound, and do the data support the conclusions?

Reviewer #1: Partly

Reviewer #2: Partly

Reviewer #3: Yes

2. Has the statistical analysis been performed appropriately and rigorously? 

Reviewer #1: No

Reviewer #2: I Don't Know

Reviewer #3: No

3. Have the authors made all data underlying the findings in their manuscript fully available?

Reviewer #1: Yes

Reviewer #2: No

Reviewer #3: Yes

4. Is the manuscript presented in an intelligible fashion and written in standard English?

Reviewer #1: Yes

Reviewer #2: Yes

Reviewer #3: No

5. Review Comments to the Author

Reviewer #1: Review of Manuscript PONE-D-23-02496: “Standing-up against moral violations: The predicting role of attribution, kinship, and severity.”

The submitted manuscript presents data investigating the effects of severity of outcome, responsibility for outcome, and kinship on self-reported willingness to intervene in response to hypothetical moral scenarios. The submitted manuscript presents findings across two studies providing support for the perceived responsibility and severity of outcome as two crucial predictors of an individual’s self-reported ratings of willingness to intervene. While this work is important, the current manuscript is difficult to parse as it relates to the stated study aims, experimental design, and analytic approach. I have outlined specific comments below.

Major Concerns:

1. The introduction states the importance of differentiating between moral courage and other forms of helping behaviors at length. As such, the ability to experimentally investigate independent effects of these hypothesized distinct forms of helping is central in the current form of the manuscript. However, in the current experimental design, moral courage is conflated with severity and is modeled in the analyses as a moderate level in the severity factor. As such, it is impossible for conclusions to be drawn regarding specificity of moral courage. It is unclear how essential this is to the study design as the hypotheses seem predominantly concerned with severity, responsibility, and kinship. If this is the case, the authors should adjust the discussion of moral courage and remove conclusions specific to moral courage and focus instead on severity of outcome.

2. Related – the lack of consistency across study 1 & 2 with regard to the “moral courage” scenario make it even more difficult to assert empirical investigation of moral courage. The authors do not provide much justification for why these violations specifically represent instances of moral courage. Furthermore, including only one scenario per category makes the specific details of the scenarios even more important as idiosyncrasies of the scenarios may be driving the effects (ie. violating political ideology vs. sexual harassment). This is stated in a roundabout way in the discussion section but should be further elaborated.

3. The authors conduct full-factorial between subjects analyses in both studies including modeling the 3-way interaction. Given that there are no hypotheses related to the three-way hypothesis and there is unlikely to be sufficient power to detect a three-way hypothesis, I would suggest the authors remove the three-way interaction from their analyses. If the authors choose to keep the three-way interaction, hypotheses and support for this inclusion should be provided in the introduction.

4. The mediation analysis in study 2 is interesting however the conclusion that it is the cost of helping underlying the effect isn’t entirely clear. The way the study is designed, one cannot rule out that it is the higher cost of NOT helping the person in need (ie. greater severity for the victim not the helper).

Minor Concerns:

1. Study 1 & Study 2 differ dramatically based on sex and age of the study population. The authors should include a comparison of the two samples across demographic variables. Study 1 in particular is ~77% female – this should be acknowledged directly and addressed in the discussion.

2. The OSF link was not functioning to see the moral scenarios

3. Lines 300-303 and 394-397 should be describing the low-severity effects but it is unclear. Please revise the language.

4. Lines 430-434 are confusingly written – is this asserting that the mediation was partial?

Reviewer #2: Comments

Abstract: Overall, the abstract effectively summarizes the main points of the research and its findings. It clearly states the research question and provides an overview of the hypotheses and predictions. However, some aspects could be further improved:

Lack of Methodological Information

Generalizability and External Validity

Limited Discussion of Implications

Incomplete Evidence on Costs of Intervention

Introduction:

The introduction lacks a clear and concise research question or hypothesis. While it mentions the aim of investigating the predicting functions of responsibility attributions, kinship, and severity on people's willingness to intervene, it does not explicitly state the specific research question that the study aims to answer. Clearly state the research question or hypothesis that the study aims to address. This will provide a clear focus for the introduction and guide the reader's understanding of the study's objectives.

The organization of the introduction is somewhat disjointed. The transition between different topics and concepts is not always smooth, and the flow of the text could be improved. Refine the organization and flow of the introduction. Ensure that there is a logical progression of ideas and that transitions between concepts are smooth.

The introduction would benefit from a clearer explanation of the theoretical frameworks (attribution theory and kinship theory) and their relevance to the study. The current explanation is brief and may leave readers wanting more context. Provide a more comprehensive explanation of the theoretical frameworks (attribution theory and kinship theory) and their applicability to the study. This will help readers understand the theoretical foundations of the research and how they relate to the specific research question.

Methodology and Results:

The study presents an investigation into the effects of attribution, situational attributes, and kinship on helping behavior. The authors provide a detailed description of their methodology, including the power analysis, participants, design, procedures, and dependent variables. The results are reported, including main effects and interactions. Overall, the study appears to be well-designed and the results are presented clearly. However, there are some areas that require further clarification and improvement. The following critical analysis addresses these issues;

Power Analysis and Sample Size: The authors state that they aimed to detect a small effect size (η² = .05) with a power of .80. However, the power analysis using GPower indicated a required sample size of 249 participants, while the actual number of participants tested was 290. This raises questions about the accuracy of the power analysis and whether the sample size was determined appropriately. The authors should provide a clear explanation or justification for the larger sample size.

Ethical Approval: The authors state that according to the University of Regensburg's ethic committee, no ethical approval was needed for the experiment. It is essential to provide more information regarding the criteria used by the ethics committee to determine that ethical approval was not necessary. This clarification is important to ensure that the study was conducted ethically and in line with ethical standards.

Random Assignment: The authors mention that participants were randomly assigned to one of the 12 experimental conditions. However, it is not clear how the randomization was performed. Providing details about the randomization procedure would enhance the transparency and rigor of the study.

Manipulation Check: The manipulation check results indicate a significant difference between high and low responsibility conditions, supporting the success of the responsibility manipulation. However, the effect size reported (d = 1.56) appears to be quite large. Such a large effect size raises concerns about the robustness of the manipulation. The authors should discuss potential limitations or alternative explanations for such a strong effect.

Marginal Effect: The authors report a marginal effect for kinship (F(1, 278) = 3.58, p = .059, ω² = .01). While the effect did not reach conventional levels of statistical significance, it is still important to interpret and discuss this finding in the context of the research question. The authors should provide a more thorough analysis and discuss potential implications of this marginal effect.

Figure 1 and Error Bars: Figure 1 presents behavioral intentions to help a person in need, but the error bars representing the 95% confidence intervals are not clearly defined. It is crucial to provide a clear explanation of the error bars, including how they were calculated and what they represent. Additionally, the figure should be labeled appropriately to provide a clear understanding of the data being presented.

Here are some deficiencies or areas that could be improved in the general Discussion section:

Lack of Quantitative Results: The section primarily focuses on summarizing the findings and interpreting them in relation to existing theories. However, it lacks specific quantitative results, such as effect sizes or statistical significance levels. Including these details would provide a more precise understanding of the findings and increase the rigor of the discussion.

Incomplete Explanation of Responsibility-Based Effects: The section mentions that responsibility-based effects were not observed in more-severe situations, but it does not delve into the possible reasons for this discrepancy. Exploring potential explanations or discussing alternative theoretical frameworks that could account for the lack of responsibility effects in severe situations would enhance the analysis.

Insufficient Explanation of Kinship Theory Findings: Although the authors acknowledge the absence of evidence for kinship theory as a predictor, the provided explanations for this discrepancy appear somewhat speculative and lack substantial empirical support. Providing a more in-depth analysis of the potential reasons for the deviation from previous research and discussing alternative explanations would strengthen this part of the discussion.

Lack of Discussion on Sample Representativeness: The section does not mention the characteristics of the sample or any potential limitations related to sample representativeness. Including information about the demographics and characteristics of the participants, as well as addressing any potential biases or limitations associated with the sample, would add depth and context to the study's findings.

Limited Generalizability of the Findings: The section does not explicitly address the generalizability of the findings beyond the specific experimental context used in the study. Recognizing the limitations of generalizability and discussing the potential boundary conditions or contexts where the findings may not hold would provide a more comprehensive understanding of the research implications.

Reviewer #3: Thank you very much for giving me this opportunity to review the manuscript entitled “Standing-up against moral violations: The predicting role of attribution, kinship, and severity.", I have a few suggestions for the author(s) to incorporate in the manuscript and improve its quality further.

1. Too complicated sentence structure that has not the capability to generalize.

2. Results are presented in too much length that must be in a concise form.

3. Basic required tests are missing.

4. Sentence is not giving complete meanings and are not well interlinked.

5. Revised the sentence of the entire document in complete, understandable, concise, and concrete form.

6. latest references need to be added.

7. The abstract and introduction are not well-written and explicitly stated.

8. What is the rationale behind using underlying methods in the current research?

9. Why the authors did not employ other similar techniques and preferred this technique solely?

10. There is a contribution of the study but is not well-written or explicitly

stated what novelty is being added to the study.

6. PLOS authors have the option to publish the peer review history of their article (what does this mean?). If published, this will include your full peer review and any attached files.

Reviewer #1: No

Reviewer #2: **Yes: **Shoaib Asim

Reviewer #3: **Yes: **Rameeza Andleeb

---

## [Author Response · Author response to Decision Letter 0]

11 Jan 2024

Editorial Comments:

E1: “PLOS ONE's style requirements” 

RESPONSE: 

Thank you for pointing this out. In accordance with PLOS ONE's style requirements, we changed the style of the headings in the manuscript and the title page’s layout. Additionally, the files are now named: Response to Reviewers, Revised Manuscript with Track Changes, and Manuscript.

E2: “Gender nouns”

Please change "female” or "male" to "woman” or "man" as appropriate, when used as a noun.

RESPONSE:

Thank you very much for this important remark. We changed all nouns according to APA’s guidelines. 

E3: “Data Availability” 

We note that you have stated that you will provide repository information for your data at acceptance. Should your manuscript be accepted for publication, we will hold it until you provide the relevant accession numbers or DOIs necessary to access your data. If you

wish to make changes to your Data Availability statement, please describe these changes in your cover letter and we will update your Data Availability statement to reflect the information you provide.

RESPONSE:

Thank you for pointing this out. As soon as the MS should be accepted, we change the OSF-repository from “View-only” to “public”. The data availability statement will then read: “All data, materials, and R-scripts are available at the OSF platform (https://osf.io/8bfe2, DOI 10.17605/OSF.IO/8BFE2) .” 

Reviewer #1: 

Major Concerns:

R1.1: “The submitted manuscript presents data investigating the effects of severity of outcome, responsibility for outcome, and kinship on self-reported willingness to intervene in response to hypothetical moral scenarios. The submitted manuscript presents findings across

two studies providing support for the perceived responsibility and severity of outcome as two crucial predictors of an individual’s self-reportedratings of willingness to intervene. While this work is important, the current manuscript is difficult to parse as it relates to the

stated study aims, experimental design, and analytic approach. I have outlined specific comments below.”

RESPONSE: 

First of all, we would like to thank you for your very helpful and constructive feedback. We address each of your comments below. 

R1.2: "The introduction states the importance of differentiating between moral courage and other forms of helping behaviors at length. As such, the ability to experimentally investigate independent effects of these hypothesized distinct forms of helping is central in the current form of the manuscript. However, in the current experimental design, moral courage is conflated with severity and is modeled in the analyses as a moderate level in the severity factor. As such, it is impossible for conclusions to be drawn regarding specificity of

moral courage. It is unclear how essential this is to the study design as the hypotheses seem predominantly concerned with severity, responsibility, and kinship. If this is the case, the authors should adjust the discussion of moral courage and remove conclusions specific to moral courage and focus instead on severity of outcome.”

RESPONSE: 

Thank you very much for this recommendation. Absolutely, we agree with you that the moral courage situations we used might be conflated with severity. However, all helping situations are, at least partly, a function of severity. Given that our pattern of results is stable, and that we used moral courage scenarios that have been successfully used in the existing literature (e.g., Fischer et al., 2006; Greitemeiyer et al., 2003, Sasse et al. 2022), we are optimistic that our findings help to better understand whether people will act morally courageous. Nonetheless, we elaborated on the potential conflation of severity (p. 21, line 495). “Nevertheless, we are aware of that moral courage situations vary in severity. Therefore, severity might interfere with the specific of moral courage situations. However, the pattern of results remained across our studies. Given that the used moral courage situations were might more severe than others, we highly encourage upcoming research to further explore whether our findings expand to less severe moral courage situations (9)“.

Moreover, we better linked our findings with moral courage wherever possible. Additionally, we shortened our explanation concerning moral courage. 

R1.3: “Related – the lack of consistency across study 1 & 2 with regard to the “moral courage” scenario make it even more difficult to assert empirical investigation of moral courage. The authors do not provide much justification for why these violations specifically represent instances of moral courage. Furthermore, including only one scenario per category makes the specific details of the scenarios even more important as idiosyncrasies of the scenarios may be driving the effects (ie. violating political ideology vs. sexual harassment). This is stated in a roundabout way in the discussion section but should be further elaborated.”

RESPONSE: 

Thank you for pointing this out. We totally agree with you that it is important to vary stimuli, especially following the Brunswikian idea of systematic and representative research designs. Given that the existing literature has shown quite stable effects for low-severe and, respectively, high-severe situations, we opted to only vary the moral courage scenario to examine whether our predictions generalize. We emphasized this argument (p. 23, line 543): “Second, we only varied the moral courage scenarios; hence our sample of prosocial situations was rather limited. Moreover, we are aware of that prosociality in moral courage is complex and can occur in different facets (69). Thus, following the Brunswikian idea of systematic and representative research designs (80–82), we highly encourage future research to systematically vary situations to get a full-fledged understanding of the predictors of prosocial behavior (i.e., moral courage).”

R1.4: “The authors conduct full-factorial between subjects analyses in both studies including modeling the 3-way interaction. Given that there are no hypotheses related to the three-way hypothesis and there is unlikely to be sufficient power to detect a three-way hypothesis, I would suggest the authors remove the three-way interaction from their analyses. If the authors choose to keep the threeway interaction, hypotheses and support for this inclusion should be provided in the introduction.” 

RESPONSE: 

Thank you very much for this remark. Indeed, we did not predict a three-way-interaction, but we included the analysis for transparency. We emphasized this in results section to make this clear for potential readers. (p. 13, line 301): “Although we did not predict a Severity by Responsibility by Kinship interaction, we included the three-way test for transparency. The three-way-interaction was not significant, F(2, 278) = 0.78, p = .457.“ The formulation for study 2 is identical. 

R1.5: “The mediation analysis in study 2 is interesting however the conclusion that it is the cost of helping underlying the effect isn’t entirely clear. The way the study is designed, one cannot rule out that it is the higher cost of NOT helping the person in need (ie. greater severity for the victim not the helper).”

RESPONSE: 

Thank you very much for this insightful remark. We agree with you that costs for not helping are a potential mediator. Given that we conceptualized our studies on previous articles, and those did not include costs for not helping a person in need, we only focused on costs for helping. Consequently, we included this highly valuable point in our limitations. Thank you again for pointing this out. (p.23, line 532). “However, our results cannot rule out the potential mediating effect of costs for not helping. For example, you observe a toddler locked in a car during summer heat. In this scenario, your costs for acting are rather low (e.g., calling 911, or smashing a side window), while the costs for not helping are extremely high. Thus, we highly encourage future research to examine costs for not helping as an additional underlying psychological process. We want to thank an anonymous reviewer for this important point.“

Minor Concerns:

R1.6: “Study 1 & Study 2 differ dramatically based on sex and age of the study population. The authors should include a comparison of the two samples across demographic variables. Study 1 in particular is ~77% female – this should be acknowledged directly and addressed

in the discussion.”

RESPONSE: 

Thank you very much for this comment. We agree with you that gender can potentially interfere with our findings (Eagly & Crowley, 1986). Consequently, we added a check for interfering effects for both studies. 

(p. 12, line 272) “Previous research revealed that men are more likely to help than women (65). Consequently, we performed a log-linear analysis to check whether participants were equally distributed across conditions regarding their gender. The model indicated no significant difference concerning participants’ gender distribution across conditions, χ2 (29) = 13.99, p = . 991.”

(p. 17, line 391) “As in Study 1, we performed a log-linear analysis to check whether participants were equally distributed across conditions regarding their gender. The model indicated no significant difference concerning participants’ gender distribution across conditions, χ2 (18) = 11.52, p = . 871.”

Regarding age, previous research revealed rather mixed evidence concerning age as a predictor for pro-social behavior (e.g., Bailey et al., 2020; McAdams et al., 1993; Steblay, 1987; Sze et al., 2012). Most results showed that children become more pro-social with increasing age (e.g., Smith & Hard, 2002). For adults, older adults seem to be more pro-social than younger (e.g., McAdams et al., 1993, Sze et al., 2012). However, the definition of older adults is rather vague; older than 37 (McAdams et al., 1993) vs 60 to 80 (Sze a., 2012). 

Nevertheless, we applied a linear model to check whether age predicted participants’ willingness to intervene. We included this additional information. 

(p. 12, line 272) “Moreover, the existing literature provides, although somewhat mixed, evidence that age can predict people’s willingness to intervene. Therefore, we ran a linear regression to check whether participants’ age predicted their willingness to intervene. The results indicate that age did not significantly predict the willingness to intervene, F(284) = 0.30, p = .581.

(p. 17, line 393) “Analogously, we ran a linear regression to check whether participants’ age predicted their willingness to intervene. The results indicate that age did not significantly predict the willingness to intervene, F(344) = 0.48, p = .490.

Moreover, the pattern of our results remained the same after including age as a control variable. To keep our results section as concise as possible, we only refer to our additional analyses that can be retrieved from OSF. 

Finally, we applied a t-test to check whether our studies differ concerning participants’ overall willingness to intervene. We added this information to our limitations. (p.23, line 548) “Third, some might argue that our samples differ concerning their demographics (i.e., age and gender), and that they are not representative. Although they might differ, we did not find a significant difference in their overall willingness to intervene, t(634) = 1.08, p = . 282, d = 0.09, 95% CI [-0.07, 0.24].”

 R1.7: “The OSF link was not functioning to see the moral scenarios.”

RESPONSE: 

Thank you for your comment and checking our material on OSF. We apologize for the inconvenience. We added a new OSF-link. You can find the moral courage scenarios in pdf named “4_mcacks_study_1_quest_eng.pdf” from p. 14 onwards. 

R1.8: “Lines 300-303 and 394-397 should be describing the low-severity effects but it is unclear. Please revise the language.

RESPONSE: 

Thank you very much for pointing this out. We rewrote both sections to improve readability. Please note that we only include one change here, because both sections are identical. 

(p 13., line 294) “However, responsibility did not significantly affect people’s willingness to intervene in moral courage situations (not responsible: M = 5.68, SD = 1.21; responsible: M = 5.41, SD = 1.45), t(278) = 1.02, p = .312. Similar results could be overserved for more-severe situation (not responsible: M = 6.17, SD = 1.25; responsible: M = 6.41, SD = 0.72, t(278) = -0.91, p = .358).”

R1.9: “Lines 430-434 are confusingly written – is this asserting that the mediation was partial?”

RESPONSE: 

Thank you very much for this remark. We now directly state that the mediation was partial. (p. 19, line 439): “However, the mediation analysis showed that costs for helping partially mediated participants’ willingness to intervene. In other words, the situation itself predicted the willingness to intervene. Precisely, participants in the moral courage condition were more willing to intervene compared to those in the less-severe helping condition (c‘1 = .0768). Moreover, participants in the more-severe condition were more willing to intervene compared to the less-severe helping condition (c‘2 = 0.457).”  

Reviewer #2: 

R2.1: “Abstract: Overall, the abstract effectively summarizes the main points of the research and its findings. It clearly states the research question and provides an overview of the hypotheses and predictions. However, some aspects could be further improved: Lack of Methodological Information, Generalizability and External Validity, Limited Discussion of Implications, Incomplete Evidence on Costs of Intervention”

RESPONSE: 

First of all, we would like to thank you for your very helpful and constructive feedback, which substantially contributed to an improvement of our manuscript. Please note that, all page and line numbers refer to the “Manuscript” file. 

Introduction

R2.2: “The introduction lacks a clear and concise research question or hypothesis. While it mentions the aim of investigating the predicting functions of responsibility attributions, kinship, and severity on people's willingness to intervene, it does not explicitly state the specific research question that the study aims to answer. Clearly state the research question or hypothesis that the study aims to address. This will provide a clear focus for the introduction and guide the reader's understanding of the study's objectives.”

RESPONSE:

Thank you very much for this remark. Our main research question was to assess whether, predictions derived from attribution and, respectively, evolutionary theory can be transferred to moral courage situations. We added this information to be clearer about our research question. (p. 7 , line 170). “In other words, we aim to answer the question whether predictions derived from attribution and, respectively, evolutionary theory can be transferred to moral courage situations.“

R2.3: “The organization of the introduction is somewhat disjointed. The transition between different topics and concepts is not always smooth, and the flow of the text could be improved. Refine the organization and flow of the introduction. Ensure that there is a logical

progression of ideas and that transitions between concepts are smooth.” 

RESPONSE:

Thank you very much for this important suggestion. We heeded your suggestion and restructured our introduction to improve its natural flow and readability. Thank you again for this advice. 

Methodology and Results

R2.4 “The study presents an investigation into the effects of attribution, situational attributes, and kinship on helping behavior. The authors provide a detailed description of their methodology, including the power analysis, participants, design, procedures, and dependent

variables. The results are reported, including main effects and interactions. Overall, the study appears to be well-designed and the results are presented clearly. However, there are some areas that require further clarification and improvement. The following critical

analysis addresses these issues”

RESPONSE:

Thank you for your important comments. We add

---

## [Decision Letter · Decision Letter 1]

19 Feb 2024

PONE-D-23-02496R1Standing-up against moral violations: The predicting role of attribution, kinship, and severity.PLOS ONE

Dear Dr. Urschler,

Thank you for submitting your manuscript to PLOS ONE. After careful consideration, we feel that it has merit but does not fully meet PLOS ONE’s publication criteria as it currently stands. Therefore, we invite you to submit a revised version of the manuscript that addresses the points raised during the review process.

Based on the comments by the reviewers, I believe that revising the manuscript will help in better understanding of the research and improve the quality.

We look forward to receiving your revised manuscript.

Kind regards,

Yasir Ahmad

Academic Editor

PLOS ONE

Journal Requirements:

Reviewers' comments:

Reviewer's Responses to Questions

**Comments to the Author**

1. If the authors have adequately addressed your comments raised in a previous round of review and you feel that this manuscript is now acceptable for publication, you may indicate that here to bypass the “Comments to the Author” section, enter your conflict of interest statement in the “Confidential to Editor” section, and submit your "Accept" recommendation.

Reviewer #1: (No Response)

Reviewer #2: (No Response)

2. Is the manuscript technically sound, and do the data support the conclusions?

Reviewer #1: Partly

Reviewer #2: Yes

3. Has the statistical analysis been performed appropriately and rigorously? 

Reviewer #1: Yes

Reviewer #2: I Don't Know

4. Have the authors made all data underlying the findings in their manuscript fully available?

Reviewer #1: Yes

Reviewer #2: (No Response)

5. Is the manuscript presented in an intelligible fashion and written in standard English?

Reviewer #1: No

Reviewer #2: (No Response)

6. Review Comments to the Author

Reviewer #1: I commend the authors on their thoughtful response to reviews, in particular their consideration of age and sex effects both within and across the studies they present and the language they add regarding important work to follow.

I do still have concerns regarding the operationalization of moral courage in the manuscript the degree to which it is conflated with severity in the analyses and the authors do not directly address this concern. It is not simply that moral courage situations are more severe than other scenarios they present but that they simultaneously posit moral courage as an outcome and as a predictor since it is is a level of the factor severity in their methodological design. In doing so, the methodological design and analytic approach conflates moral courage with severity such that the relationship between them cannot be empirically tested. As I stated in my original review - I don't see this as a major issue with the analyses presented here but rather with the language/conclusions drawn by the authors. In the current version of the manuscript it is unclear whether the authors posit that varying levels of severity of a situation are important mechanisms for the prediction of moral courage or whether moral courage represents a specific level of severity.

Finally, I noticed some issues with wording/typos in the responses that impacted the clarity of the responses. A close review of the language throughout the manuscript and edits for clarity when typos are present is needed. For example (typos/missing words highlighted with ** below):

"Therefore, severity might interfere with the **specific of** moral courage situations. However, the pattern of results remained ** ** across our studies. Given that the used moral courage situations **were might** more severe than others, we highly encourage upcoming research to further explore whether our findings expand to less severe moral courage situations "

"Moreover, we are aware **of that** prosociality in moral courage is complex and can occur in different facets "

Reviewer #2: Introduction

The introduction provides a good overview distinguishing moral courage from helping behavior. However, it would be helpful to more clearly state the purpose and hypotheses of the current studies early on, rather than waiting until the end of the introduction.

When introducing the key predictions, explain the rationale more clearly about why responsibility attributions and kinship would interact with situation severity. Elaborate on the underlying theories and logic so the reader understands why these factors are expected to function differently across less severe vs more severe situations.

Study 1

In the discussion of Study 1 limitations, clarify whether all participants actually had a sibling, since assigning some only children to a "sibling condition" could create issues. Recommend changing the terminology to "close relative" in Study 2.

Provide more interpretation of the marginal kinship effect in the results and whether it suggests the predicted relationship.

Explain why moral courage may still be less severe than other emergencies and whether the findings could extend to less severe moral courage situations.

Study 2

The hypothesis section could state more directly that Study 2 aims to replicate Study 1 and investigate perceived costs as a mediator. Make the goals clear upfront.

Report the specific items used to measure perceived costs and provide reliability for this scale.

In the mediation analysis, explain the meaning of the direct effect results for severity on helping intentions. Does this suggest partial vs full mediation?

General Discussion

Summarize clearly which hypotheses were and were not supported overall across the two studies.

Provide more interpretation about why kinship effects were not found and potential reasons or moderators to investigate in future research.

Discuss whether the findings for responsibility attributions may extend to less severe moral courage situations, given the limitations noted about severity.

Conclusions could identify practical implications about encouraging moral courage and helping behaviors.

7. PLOS authors have the option to publish the peer review history of their article (what does this mean?). If published, this will include your full peer review and any attached files.

Reviewer #1: No

Reviewer #2: **Yes: **Dr. Shoaib Asim

---

## [Author Response · Author response to Decision Letter 1]

30 Apr 2024

Editorial Comments:

E1: “Reference list” 

Please review your reference list to ensure that it is complete and correct.

RESPONSE:

Thank you for pointing this out. We again checked our reference list. Our refence list is complete, correct, and does not include any retracted references. 

 

Reviewer #1: 

R1.1: “I commend the authors on their thoughtful response to reviews, in particular their consideration of age and sex effects both within and across the studies they present and the language they add regarding important work to follow.”

RESPONSE: 

First of all, we would like to thank you again for your very helpful and constructive feedback. We address each of your comments below. 

R1.2: " I do still have concerns regarding the operationalization of moral courage in the manuscript the degree to which it is conflated with severity in the analyses and the authors do not directly address this concern. It is not simply that moral courage situations are more severe than other scenarios they present but that they simultaneously posit moral courage as an outcome and as a predictor since it is is a level of the factor severity in their methodological design. In doing so, the methodological design and analytic approach conflates moral courage with severity such that the relationship between them cannot be empirically tested. As I stated in my original review - I don’t see this as a major issue with the analyses presented here but rather with the language/conclusions drawn by the authors. In the current version of the manuscript it is unclear whether the authors posit that varying levels of severity of a situation are important mechanisms for the prediction of moral courage or whether moral courage represents a specific level of severity.”

RESPONSE: 

Thank you very much for pointing this out. Although we argue that moral courage intuitively refers to a rather high level of severity, we agree that moral courage scenarios come in different facets (Sasse et al., 2022). Especially, moral courage scenarios do vary in severity. The scenarios that we used in our MS are rather high in severity. However, we acknowledge that there are moral courage situations that might be less severe than the scenarios that we used, but still more severe than most helping scenarios. For example, speaking-up against an inappropriate joke might be less severe for an intervener than preventing someone from a physical attack (Study 1) or sexual harassment (Study 2), although more severe than lending class notes, or picking up coins, etc. Consequently, we made the point clearer in the general discussion. Additionally, we added that future research should also include full designs with moral courage/helping situations crossed by severity (less, more) to disentangle the separate effects of type of situation and severity. Thank you again for your feedback, which has helped to improve the clarity of our argument. 

The section now reads (p. 21, line 474)” ). To be clear, moral courage indicates a higher level of severity.”

(p23. line 547)” Moreover, we are aware that prosocial behavior in moral courage is complex and can occur in different facets. Especially, moral courage scenarios do vary in severity (9). For example, speaking-up against an inappropriate joke might be less severe for an intervener than preventing someone from a physical attack (Study 1) or sexual harassment (Study 2), although more severe than lending class notes, or picking up coins, etc. Thus, following the Brunswikian idea of systematic and representative research designs (77–79), we highly encourage future research to systematically vary situations to get a full-fledged understanding of the predictors of prosocial behavior (i.e., moral courage). Particularly, upcoming research should orthogonally manipulate severity in moral courage scenarios.“

R1.3: “Finally, I noticed some issues with wording/typos in the responses that impacted the clarity of the responses. A close review of the language throughout the manuscript and edits for clarity when typos are present is needed. For example (typos/missing words highlighted with ** below):”

RESPONSE: 

Thank you for your dedication to our MS. We do apologize for these inconveniences. Accordingly, we fixed our typos, added the missing words, and let proofread our MS. Thank you again!

 

Reviewer #2: 

R2.1: “The introduction provides a good overview distinguishing moral courage from helping behavior. However, it would be helpful to more clearly state the purpose and hypotheses of the current studies early on, rather than waiting until the end of the introduction.”

RESPONSE: 

First of all, we would like to thank you again for your feedback, which improved our manuscript. Please note that, all page and line numbers refer to the “Manuscript” file. We now state the purpose and hypotheses very early in the introduction. 

R2.2: “When introducing the key predictions, explain the rationale more clearly about why responsibility attributions and kinship would interact with situation severity.”

RESPONSE:

Thank you for this comment. We added further information about our rationale of the predicted interaction (p. 7, line 166/167) “Therefore, little is known whether these findings to generalize to moral courage scenarios.“

R2.3: “Elaborate on the underlying theories and logic so the reader understands why these factors are expected to function differently across less severe vs more severe situations.” 

RESPONSE:

Thank you for this suggestion. We elaborated on why severity should affect people’s willingness to intervene. 

(p. 8, line 195) “In less-severe situations, people’s willingness to intervene is predicted to decrease, when the receiver of help is perceived to be responsible for her plight, because failing to help would have little consequences. However, we predict that people’s willingness to intervene increases when they perceive the receivers of help as being not responsible (14,16).“

(p. 9, 202) “Furthermore, we predict an opposed interaction effect between kinship and situation’s severity: in more-severe situations, people’s willingness to help a relative is higher, compared to a non-relative, but no significant effect of relationship is predicted for less-severe situations, because helping increase ultimately the prevalence of a helper’s genes (23,54).”

R2.4 “In the discussion of Study 1 limitations, clarify whether all participants actually had a sibling, since assigning some only children to a “sibling condition” could create issues. Recommend changing the terminology “close relative” in Study 2.”

RESPONSE:

We very much agree with this comment and corrected accordingly this wording in the first revision of our MS. Moreover, we have used the term “close relative” in study 2 since the initial submission and now made sure that this is done consistently.

R2.5 “Provide more interpretation of the marginal kinship effect in the results and whether it suggests the predicted relationship.”

RESPONSE:

Thank you for this remark. We included additional explanations in the previous version of our MS. In total, we provide five potential explanations why our studies did not show kinship effects. We derived our explanations from established theories and existing findings that are related to our work. However, we made sure that the different explanations are clearly represented. Nevertheless, we leave it up to the editor whether it’s crucial to add further explanations. 

R2.6 “Explain why moral courage may still be less severe than other emergencies and whether the findings could extend to less severe moral courage situations.”

RESPONSE:

We argue that moral courage represents a rather high level of severity. We are aware of that moral courage scenarios come in different facets (Sasse et al., 2022). Especially, moral courage scenarios do vary in severity. The scenarios that we used in our MS are rather high in severity. However, we acknowledge that there are moral courage situations that might be less severe than the scenarios that we used, but still more severe than most helping scenarios. For example, speaking-up against an inappropriate joke might be less severe for an intervener than preventing someone from a physical attack (Study 1) or sexual harassment (Study 2), although more severe than lending class notes, or picking up coins, etc. Consequently, we made to point clearer in we the general discussion. Additionally, we added that future research should also include moral courage situations that are less severe. 

The section now reads (p. 23, line 547) ”Moreover, we are aware that prosocial behavior in moral courage is complex and can occur in different facets. Especially, moral courage scenarios do vary in severity (9). For example, speaking-up against an inappropriate joke might be less severe for an intervener than preventing someone from a physical attack (Study 1) or sexual harassment (Study 2), although more severe than lending class notes, or picking up coins, etc. Thus, following the Brunswikian idea of systematic and representative research designs (77–79), we highly encourage future research to systematically vary situations to get a full-fledged understanding of the predictors of prosocial behavior (i.e., moral courage). Particularly, upcoming research should orthogonally manipulate severity in moral courage scenarios.“

R2.7 “The hypothesis section could state more directly that Study 2 aims to replicate Study 1 and investigate perceived costs as a mediator. Make the goals clear upfront.“

RESPONSE:

Thank you for pointing this out. In line with your recommendation, the first sentence of Study 2 now reads: “Study 2 aimed to replicate Study 1, and to examine costs for helping as an underlying psychological process.

R2.8: “Report the specific items used to measure perceived costs and provide reliability for this scale.”

RESPONSE:

We agree with this comment as this information was already included in the first version of our MS. 

R2.9: “In the mediation analysis, explain the meaning of the direct effect results for severity on helping intentions. Does this suggest partial vs full mediation?“ 

RESPONSE:

Again, we appreciate this comment and checked that the information about the mediation that we included in the revised manuscript is clearly represented.

R2.10: “Provide more interpretation about why kinship effects were not found and potential reasons or moderators to investigate in future research.” 

RESPONSE:

Please see response to comment R2.5

R2.11: “Discuss whether the findings for responsibility attributions may extend to less severe moral courage situations, given the limitations noted about severity.” 

RESPONSE:

Thank you for pointing this out. Please see our response R2.6

R2.12: “Conclusions could identify practical implications about encouraging moral courage and helping behaviors.”

RESPONSE:

Thank you for your remark. We added a potential practical implication of our findings (p. 24, line 569). “In addition to our theoretical contributions, our findings can be applied to improve training programs that foster prosocial behavior. Especially, our results can help to develop tailored exercises that aim to improve moral courage.”

---

## [Decision Letter · Decision Letter 2]

11 Jul 2024

Standing-up against moral violations: The predicting role of attribution, kinship, and severity.

PONE-D-23-02496R2

Dear Dr. Urshchler,

We’re pleased to inform you that your manuscript has been judged scientifically suitable for publication and will be formally accepted for publication once it meets all outstanding technical requirements.

Kind regards,

Yasir Ahmad

Academic Editor

PLOS ONE

Additional Editor Comments (optional):

Reviewers' comments:

Reviewer's Responses to Questions

**Comments to the Author**

1. If the authors have adequately addressed your comments raised in a previous round of review and you feel that this manuscript is now acceptable for publication, you may indicate that here to bypass the “Comments to the Author” section, enter your conflict of interest statement in the “Confidential to Editor” section, and submit your "Accept" recommendation.

Reviewer #1: All comments have been addressed

Reviewer #2: All comments have been addressed

2. Is the manuscript technically sound, and do the data support the conclusions?

Reviewer #1: Yes

Reviewer #2: Partly

3. Has the statistical analysis been performed appropriately and rigorously? 

Reviewer #1: Yes

Reviewer #2: Yes

4. Have the authors made all data underlying the findings in their manuscript fully available?

Reviewer #1: Yes

Reviewer #2: Yes

5. Is the manuscript presented in an intelligible fashion and written in standard English?

Reviewer #1: Yes

Reviewer #2: Yes

6. Review Comments to the Author

Reviewer #1: Thank you for your continued responsiveness to my comments. All of my comments and concerns have been adequately addressed!

Reviewer #2: I review suggestions and questions, then check that all comments have been addressed in manuscript.

7. PLOS authors have the option to publish the peer review history of their article (what does this mean?). If published, this will include your full peer review and any attached files.

Reviewer #1: No

Reviewer #2: **Yes: **Dr. Shoaib Asim

---

## [Editor Report · Acceptance letter]

1 Aug 2024

PONE-D-23-02496R2 

PLOS ONE

Dear Dr. Urschler, 

I'm pleased to inform you that your manuscript has been deemed suitable for publication in PLOS ONE. Congratulations! Your manuscript is now being handed over to our production team.

Kind regards, 

on behalf of

Dr. Yasir Ahmad 

Academic Editor

PLOS ONE